# Metasurface enabled broadband all optical edge detection in visible frequencies

Ibrahim Tanriover [1], Sina Abedini Dereshgi [1] & Koray Aydin [1]

Image processing is of fundamental importance for numerous modern technologies. In recent years, due to increasing demand for real-time and continuous data processing, metamaterial and metasurface based all-optical computation techniques emerged as a promising alternative to digital computation. Most of the pioneer research focused on all-optical edge detection as a fundamental step of image processing. Metasurfaces have been shown to enable real time edge detection with low to no power consumption. However, the previous demonstrations were subjected to the several limitations such as need for oblique-incidence, polarization dependence, need for additional polarizers, narrow operation bandwidth, being limited with processing in 1D, operation with coherent light only, and requiring digital post-processing. Here, we propose and experimentally demonstrate 2D isotropic, polarization-independent, broadband edge detection with high transmission efficiency under both coherent and incoherent illumination along the visible frequency range using a metasurface based on Fourier optics principles.

Image processing techniques play an increasingly significant role in several emerging technologies such as autonomous driving[1], biometric verification[2], and AR/VR glasses[3]. Digital computers have long performed these computation tasks. However, their computational cost started to become a limiting factor with the increasing demand for real-time, continuous data processing. Optical computation techniques emerged as an alternative solution with their ability to provide ultrafast image processing with negligible power consumption and to allow parallel processing. Although conventional bulky optics suffer from large footprint and difficulties in integration, nanophotonic approaches based on metamaterials and metasurfaces have proven to successfully address such problems. In recent years, several approaches based on metamaterials and metasurfaces have been quite successful in performing analog computation, specifically, edge detection tasks. Earlier work put emphasis on the Fourier optics principles based on 4f systems, where lenses are used to perform spatial Fourier transformations of the optical fields creating 4f distance between the object and the image[4–6]. In addition to numerous theoretical proposals[7–12], optical computing based on Pancharatnam-Berry phase[13, 14] and tailored resonances[15] have been demonstrated experimentally. However, both of these approaches have certain limitations. The former approach requires linear input polarization together with

additional polarizers at the input and output ports of the system and the latter example is limited with narrow-band operation and works in the reflection mode.

More recently, a more compact approach of using metasurfaces for optical data processing emerged. In this approach, data processing is facilitated by engineering the angle-dependent optical response of the metasurface, which corresponds to filtering in the momentum (**k**) space without the need for a Fourier transformation[5,6,16–32]. Despite offering size-reduction in the optical system, this method requires device-specific angle[5,20,24,26–28,30], polarization dependency[16,22], and coherent illumination[5,17], and operates over narrow bandwidth[5,16,17,19,23–26,29,30], with limited numerical aperture (NA)[5,20,21,23,25,28]. All these limitations prevent the applicability of such platforms in real-life applications such as augmented reality and self-driving cars. Although recent work attempted to address the limitations on NA and operation bandwidth[5,19,32], they are limited to coherent laser sources. The coherence requirement was addressed whereas it requires digital post-processing, which significantly reduces the cost advantage of analog computation[17]. Considering modern-day applications, there is an utmost need for polarization-insensitive, broadband, and 2D all-optical image processing devices operating along the visible

[1]Department of Electrical and Computer Engineering, Northwestern University, Evanston, IL 60208, USA. ✉e-mail: aydin@northwestern.edu

wavelength range under both coherent and incoherent illuminations.

In this work, simultaneously satisfying all listed requirements and addressing aforementioned challenges, we propose and experimentally demonstrate a sub-wavelength metallic metasurface that can perform all-optical edge detection, a fundamental step in many image processing tasks. Our proposed metasurface device operates in the transmission mode that is desired for several real-life applications. In designing our metasurface-based edge-detection platform, we rely on the 4f approach that offers ease of integration with conventional or compact meta-optical imaging and sensing systems. In our experiments, we demonstrated edge detection under various illumination conditions; coherent illumination with red and green lasers, with linear and random polarization states, and incoherent, unpolarized, broadband illumination from a light bulb mimicking the real-life scenario. Additional simulations and measurements were performed in the near-IR range confirming edge detection over a broader wavelength range exceeding the optimized operation band.

## Results

### Target response and unit cell design

Edge detection can be implemented with a 4f system through spatial filtering. The 4f system employs lenses to carry out spatial Fourier transformations of the electromagnetic fields. In this system, the object is located at the front focal plane of the first lens. The spatial filter, which has the required transfer function, is located at the back focal plane of the first lens. This plane also serves as the front focal plane of the second lens. The image is formed at the back focal plane of the second lens creating a 4f distance between the object and the image, where f is the focal length of the lenses. The transfer function of the spatial filter corresponds to the Fourier transform of the mathematical operation applied to the image[4–6]. Conventionally edge detection is achieved using sharp high-pass spatial filters[31], which have a trivial transfer function in the ray optics regime. Mathematically, another method for edge detection is spatial differentiation, which can also be implemented through 4 f systems as a linear space invariant

operation. This method provides better edge detection quality (larger signal-to-noise ratio) than conventional high-pass filtering as shown in Fig. S2. However, the transfer function is not as trivial as the former. Considering the simple case of 1D operation, the $n$th order differentiator $h(x) = \frac{\partial^n}{\partial x^n}$ has a transfer function $H(k_x) = (ik_x)^n$[4–6,15]. At the Fourier plane (where the differentiator is located), such transfer functions can be calculated by applying $k_x \to \frac{x}{R}$ transformation, where $-R \leq x \leq R$ and $2R$ is the device size. For 2nd order differentiation, corresponding spatial transmission response becomes $t(x) = -(\frac{x}{R})^2$.

To realize the desired optical response, we created a library of 2D unit cells with a fixed periodicity of 500 nm. The unit cells consist of Aluminum wires with a constant height and varying widths ranging from 0 to 500 nm on top of a SiO$_2$ substrate (Fig. 1a). A width of 0 nm corresponds to a bare substrate, while 500 nm represents a continuous film. The height of the metal stripe is set to 70 nm to ensure no transmission in the 500 nm width (continuous film case). The minimum stripe width (excluding the no stripe case) was set to 50 nm due to fabrication constraints.

We used FDTD Lumerical software to simulate the optical response of the unit cell library (see methods section for simulation details). The results, shown in Fig. 1b, indicate that the amplitude of the transmitted light gradually increases from 0 to 1 as the width of Al stripe decreases, which satisfies the transmission response requirements for the target response. Moreover, the amplitude response (Transmission, T) exhibits no significant spectral deviations, particularly in the 500 nm to 800 nm wavelength range, indicating that a selected set of unit cells can reasonably approximate the target response over a broad spectral range.

For the 2nd order differentiator, the phase requirement is that it be constant, with small deviations being acceptable. Broadband operation is not dependent on phase dispersion and can be achieved as long as the required constant phase response is satisfied at each frequency point. As seen in Fig. 1c, the phase response ($\varphi$) largely fulfills this requirement.

In order to design a hypothetical device with a width of 50 μm, we chose a set of unit cells from the design library by fitting their optical

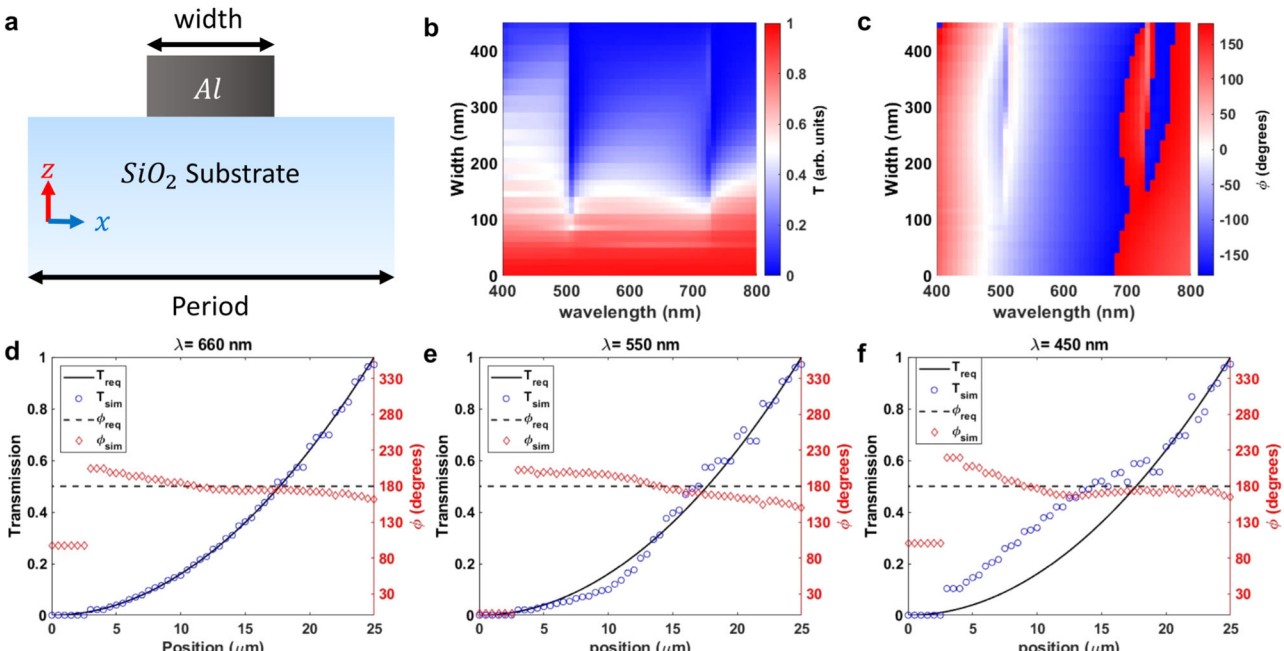

**Fig. 1 | Unit cell simulations. a** The unit cell structure: an Aluminum ridge on top of a SiO$_2$ substrate. The periodicity is 500 nm and height of the Al layer is 70 nm. The width varies between 0 nm to 500 nm. **b** Transmission and (**c**) phase response for

varying width from 400 nm to 800 nm wavelengths. The required versus simulated optical response of the selected set of unit cells at (**d**) 660 nm, (**e**) 550 nm, and (**f**) 450 nm wavelengths.

response at 660 nm to the target response ($t(x) = -\left(\frac{x}{R}\right)^2$; $R = 25\,\mu m$). As shown in Fig. 1d, the unit cell library provides an almost perfect approximation to the target amplitude, and the phase response exhibits deviations within $\pm 20°$ range. It is worth noting that the phase response of the no transmission case was not considered since it does not contribute to the optical response of the device. The selected unit cell set (or the hypothetical device) maintains a reasonable approximation to the target response at 550 nm, in terms of both phase and transmission (Fig. 1e). However, the deviation in transmission becomes substantial at 450 nm (Fig. 1f). Note that the metasurface still maintains a gradually increasing transmission profile from center towards the edge. It suppresses the low-frequency components and operates as a high-pass spatial filter, which is desired for edge detection. As a result, our metasurface is expected to maintain edge detection operation at 450 nm with a relatively lower quality than the design wavelength. Same unit cell library was also simulated in near-IR range, and the results are provided in Supplementary Information Fig. S3.

## 3D device design and simulations

We map the 2D hypothetical device to 3D by converting position to radial distance ($x \rightarrow r$; $r = \sqrt{x^2 + y^2}$). This is basically a coordinate system transformation from cartesian to cylindrical coordinates. In this case, the underlying infinity assumption for the in-plane orthogonal dimension of the unit structure, i.e., stripe length in $y \rightarrow \infty$, in 2D simulations maps to the $\Phi$ (azimuthal) component for the 3D device. Since the azimuthal axis has a range from 0 to $2\pi$, the stripe length becomes $2\pi$ in $\Phi$ corresponding to a full circle. Hence, the resulting device becomes a collection of co-centric rings of Al ridges with varying width, as schematically depicted in Fig. 2a, e. The cylindrical symmetry of the metasurface ensures isotropic and polarization-independent operation.

The cylindrical metasurface is simulated both at visible and near-IR wavelength range using unpolarized excitation incident from the substrate side. Figure 2b–d shows the amplitude of the transmitted electric field's spatial distribution and Fig. 2f–h shows the

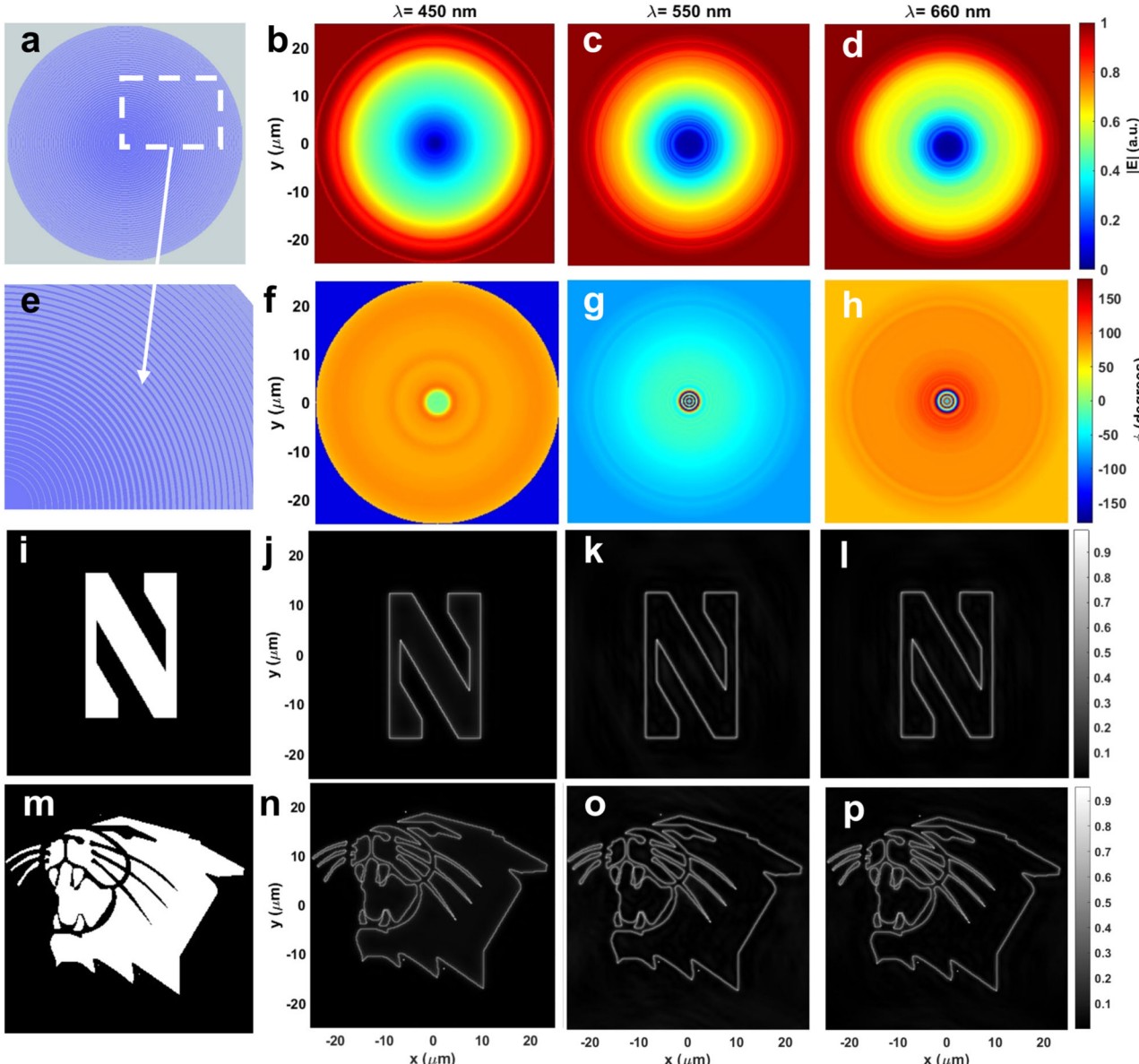

**Fig. 2 | 3D device simulations in visible wavelengths. a** The designed complete layout and (**e**) a closer image of the 3D metasurface: co-centric Aluminum rings of 70 nm fixed height and varying width on top of a SiO2 substrate. **b–d** The simulated amplitude and (**f–h**) phase maps of the transmitted electric field at 450, 550, and 660 nm wavelengths using the hypothetical device shown in (**a**). **i** The letter "N" of Northwestern University logo and (**m**) the Northwestern Wildcats logo as the input fields. **j–l**, **n–p** The calculated output fields of the 4f system with the metasurface.

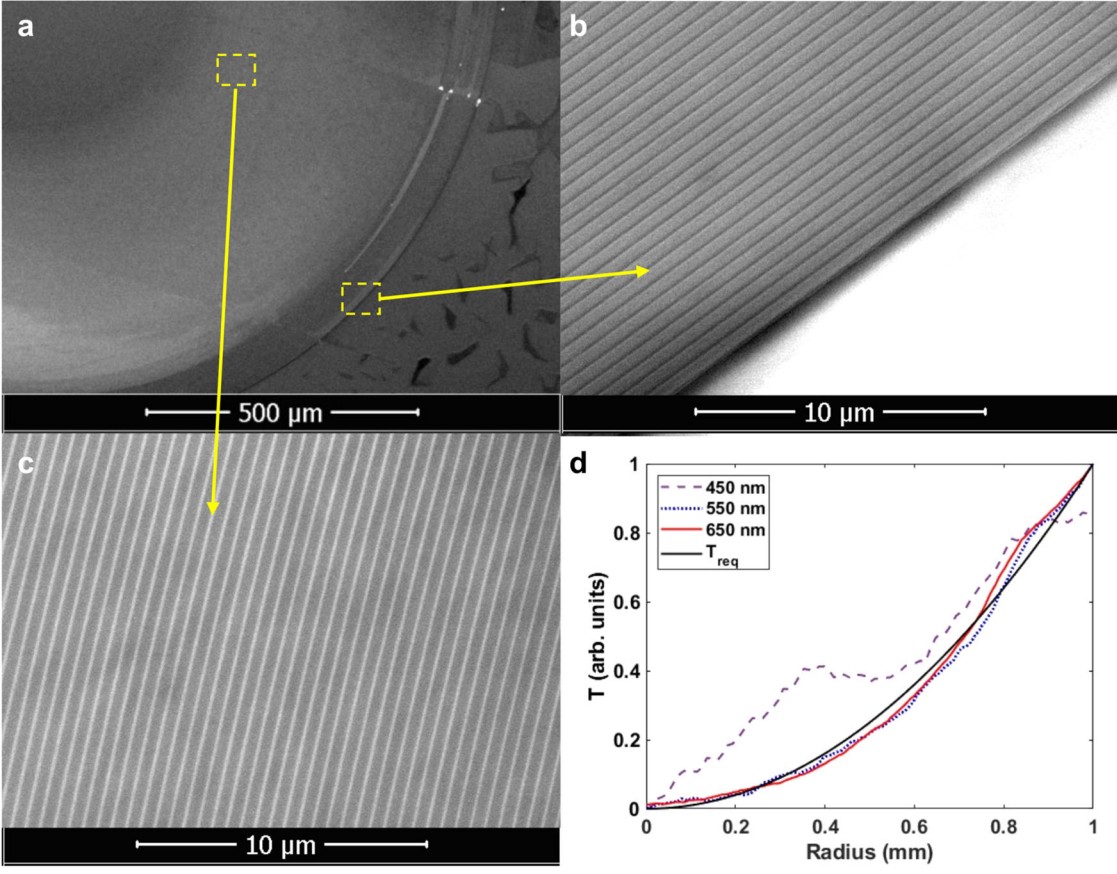

**Fig. 3 | SEM images and transmission profile of the metasurface. a–c** SEM images of the metasurface. **d** The target transmission profile (Treq) (black line) and transmission profile of the metasurface along its radius at 450 nm (purple dashed line), 550 nm (blue dotted line), and 660 nm (red line) wavelengths.

corresponding phase distribution at 450 nm, 550 nm, and 660 nm wavelengths, respectively. As seen in these figures, the optical response of 3D device is in perfect agreement with that of 2D simulations. Simulation results for 400 nm, 700 nm, and near-IR wavelengths from 800 nm to 1600 nm are provided in the Supplementary Information Figs. S4, S5.

To validate edge-detection capability of the proposed cylindrically symmetric metasurface device, two test images are selected as input fields; the letter "N" of the Northwestern University logo and the Northwestern Wildcats logo. The output of the 4f system, with the metasurface serving as the spatial filter, is numerically calculated, and resulting images are shown in Fig. 2j–l, n–p. It is clear from these simulated output images that the proposed metasurface spatial filter successfully performs edge detection operation in 2D with unpolarized coherent excitation at red, green, and blue wavelengths (660 nm, 550 nm, and 450 nm), while suppressing the background. Although the response function of the device deviates from the target response at 450 nm (Fig. 1f and Fig. 2b, f), the metasurface maintains its edge detection ability as it still operates as a high-pass filter blocking low spatial frequencies. However, since its transmission is larger than the target response at low special frequencies (closer to center of device), the background suppression is weaker at 450 nm than 550 and 660 nm wavelengths.

## Sample fabrication and characterization

To experimentally verify the simulation results, we designed and fabricated a metasurface consisting of co-centric Al rings (Fig. 2a). The metasurface diameter is set to 2 mm. The widths of individual Al rings are chosen using the results obtained from 2D simulations (Fig. 2b, c) at 660 nm wavelength. The minimum feature size (Al ring width or

distance between two Al rings) is limited to 50 nm due to fabrication constraints. The metasurface is fabricated using e-beam lithography followed by metal deposition and lift-off. Scanning electron microscope (SEM) is performed to characterize fabricated metasurface and the SEM images of the device are shown in Fig. 3a–c. SEM images with higher magnification are provided in the Supplementary Information Fig. S6.

The optical response of the metasurface is measured using an inverted microscope-spectrometer setup. The spatial transmission profiles of the metasurface along its radius for different wavelengths are plotted in Fig. 3d. The measured spatial transmission profile at 550 nm and 660 nm wavelengths are in good agreement with the simulated target response ($T_{req}$), while the transmission profile at 450 nm deviates from the target. At the three spectral points of consideration, the measurements are in great agreement with the simulations shown in Fig. 1. At 450 nm wavelength, since the transmission amplitude increases from 0 to ~0.8 towards the periphery, the metasurface still acts as a high-pass spatial filter. Thus, we expect our edge-detection device to function at a blue wavelength range of the visible spectrum as it does in the simulations (Fig. 2j, n). We also measured the transmission profiles at near-IR and provided the results in the Supplementary Information Figure S7.

## Edge detection experiments

To experimentally verify the edge-detection operation, we prepared a 4f setup using conventional lenses shown in Fig. 4a. An amplitude mask is placed at the object plane ($z_i = 0$) and illuminated coherently with the laser sources placed behind it (source distance, $|z_s| \gg f$). The CCD camera is placed at the image plane ($z_o = 4f$) to record the output field. The metasurface is placed at the Fourier plane ($z_f = 2f$). For

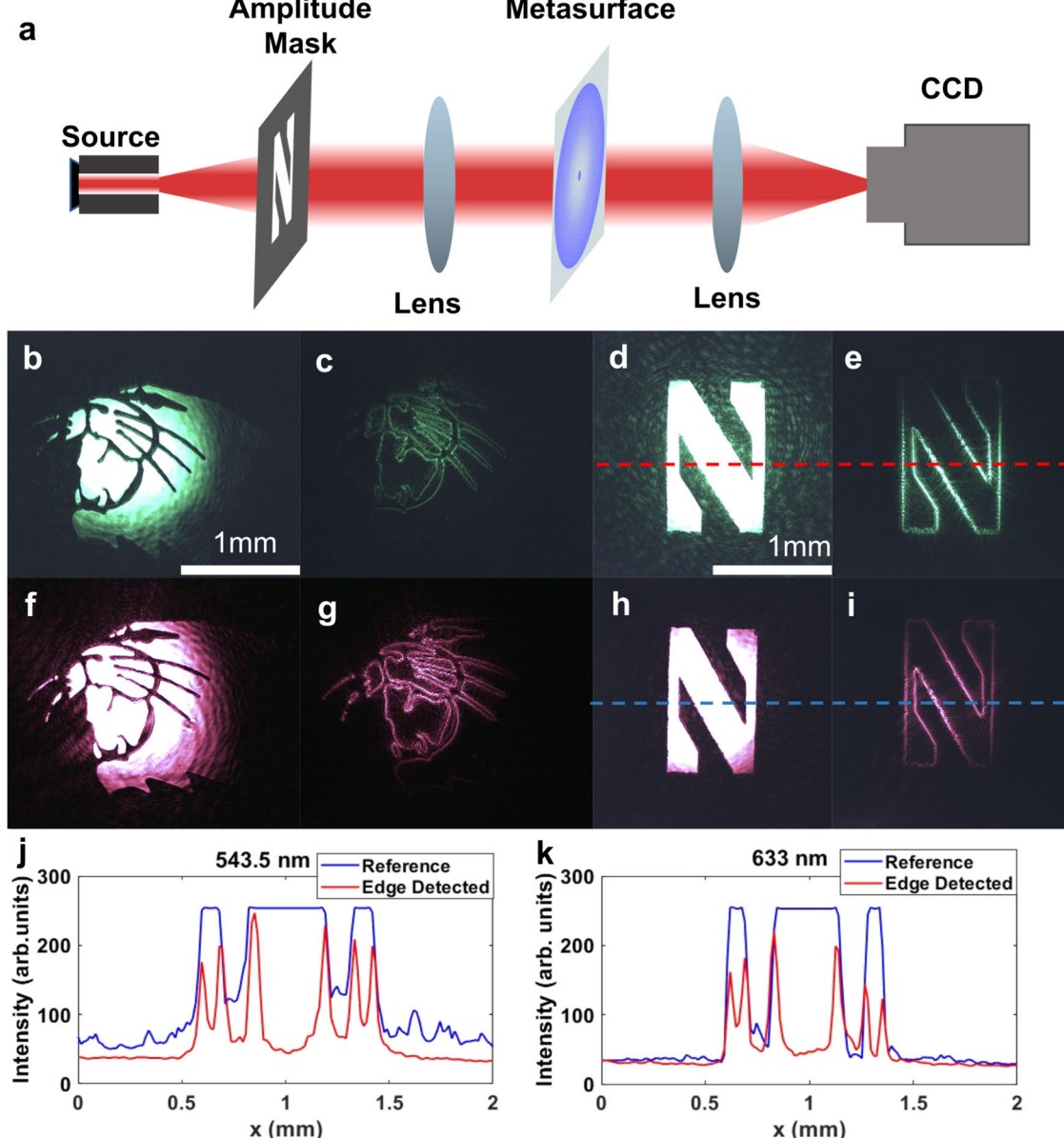

**Fig. 4 | Edge-detection experiments with Red and Green HeNe laser sources.**
**a** The schematic of the experimental setup for imaging and edge detection
experiments. **b–e** Experiments at 543.5 nm wavelength (illuminated with XYZ green
laser). **b, d** The reference (no metasurface) and (**c, e**) edge detected (with meta-
surface) images of (**b, c**) the Northwestern Wildcats logo and (**d, e**) the letter "N" of
Northwestern University logo. **f–i** Experiments at 633 nm wavelength (illuminated
with HeNe red laser). **f, h** The reference (no metasurface) and (**g, i**) edge detected
(with metasurface) images of (**f, g**) the Northwestern Wildcats logo and (**f, h**) the
letter "N" of Northwestern University logo. **j, k** The horizontal cuts of the images
under (**j**) green and (**k**) red laser illumination. The dashed lines in (**d**), (**e**), (**h**), and (**i**)
mark the location of the horizontal cuts.

reference measurements metasurface is removed. To achieve the same
input fields as the simulations, the letter "N" of the Northwestern
University logo and the Northwestern Wildcats logo, we prepare two
amplitude masks by laser cutting the corresponding images out of a
15 mm thick molybdenum sheet. Randomly polarized green (543.5 nm)
and linearly polarized red (633 nm) HeNe lasers are used as excitation
sources. The reference images (without the metasurface) at 543.5 nm
and 633 nm wavelengths are shown in Fig. 4b, d and Fig. 4f, h,
respectively. The edge detected images (with the metasurface) at
543.5 nm and 633 nm wavelengths are shown in Fig. 4c, e and Fig. 4g, i,
respectively. As clearly seen in the images obtained with the meta-
surface spatial filter, edge detection is achieved at both 543.5 nm and
633 nm wavelengths. Edge detection from randomly polarized green
laser experimentally confirms polarization-insensitive operation of the
metasurface, which is expected due to inherent cylindrical symmetry.

The edge quality and the background suppression can be further
investigated from the horizontal cross-section line-cuts of the images
that are provided in Fig. 4j, k.

To confirm that our metasurface simultaneously satisfies the tar-
get requirements on bandwidth, polarization, isotropy, and coher-
ence, we replaced the laser source with a Xenon arc lamp in the 4f
system (Fig. 4a) since its broadband output spectrum is continuous
and almost uniform across the visible region. The Xenon lamp pro-
duces an unpolarized and incoherent image at the input with an area of
coherence diameter of 8λ. It is worth noting that this value is
approximately 34λ for the sunlight according to van Cittert-Zernike
theorem[33]. As a result, it represents a close approximation to real-life
scenario applications, such as autonomous driving, where the input
field will be unpolarized, incoherent, and broadband. The metasurface
was tested with the same amplitude masks. The reference images

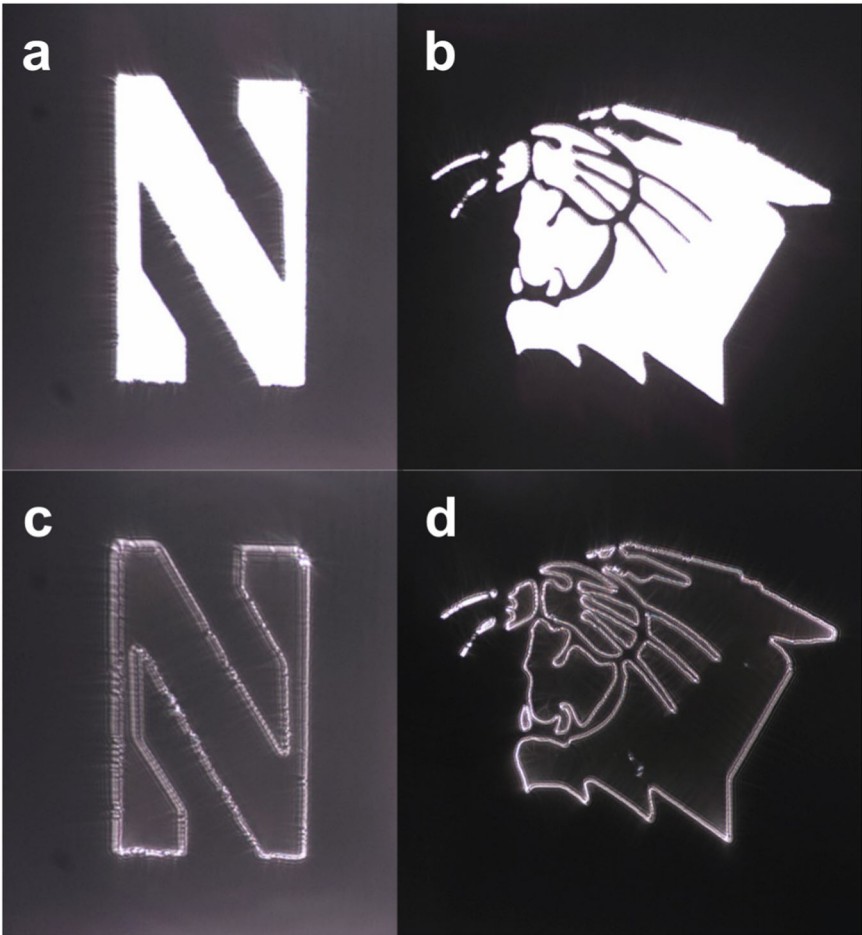

**Fig. 5 | Edge detection experiments with Xe light bulb as the light source. a, b** The reference (no metasurface) and (**c, d**) edge detected (with metasurface) images of (**a, c**) the letter "N" of Northwestern University logo and (b,d)the Northwestern Wildcats logo.

(without the metasurface) are shown in Fig. 5a, b, and the edge detected images (with the metasurface) are provided in Fig. 5c, d. As seen in Fig. 5, isotropic edge detection is achieved in the case of incoherent, unpolarized, and broadband input fields that cover the entire visible region. Additionally, we provide a supplementary video demonstrating real-time edge detection as the metasurface and the amplitude masks are aligned to the same optical path (Supplementary Video 1, 2).

To date, various edge-detection efficiency definitions are reported in literature[13, 14,22,32], and there is not an agreement on the performance metrics. Here, we propose to use contrast as an alternative figure-of-merit, which measures the difference between bright and dark regions in an image. Assuming perception by CCD instead of human eye, we replace the relative luminescence with the geometric average of intensity at the conventional definition. The resulting formula becomes contrast = $\frac{\max(I) - \min(I)}{I_{range}}$, where $I$ is the image intensity and the $I_{range}$ is the available range of the CCD. Based on this formulation, high contrast is a desirable property. For example, in the case of postprocessing of an edge detected image, to distinguish between the bright edges and dark background, larger threshold values can be applied in the higher contrast images than the lower contrast ones. Larger thresholds, in turn, provide better immunity against noise and decrease error rate. Contrast is calculated for low intensity images [Supplementary Figure S8] to avoid saturation. The calculated contrast values are 0.95 and 0.45 for the reference and edge detected images, respectively.

To further assess the quality of edge detection, we calculated the efficiency metrics proposed by Cotrufo et al.[32]. As in the contrast

calculations, to avoid detector saturation at the reference, the calculations are performed for a low intensity case shown in Supplementary Information Fig. S8. The peak efficiency, $\eta_{peak} = \max(I_{out})/\max(I_{in})$, is calculated as 48%. And, the average efficiency, $\eta_{avg} = \mathrm{avg}(I_{out}^{edge})/\mathrm{avg}(I_{in})$, is calculated as 37%. Note that the peak efficiency is in a good agreement with the ratio of the contrasts of edge detected image and reference image (~1/2). The efficiency values calculated here are an order of magnitude higher than the values reported by Cotrufo et al.[32]. We attribute this difference to the high NA of our metasurface. As it is operating at the Fourier plane, our metasurface can satisfy the target response for the spatial frequencies from 0 to 1, as seen in simulations (Fig. 1) and measurements (Fig. 3), which maximizes the efficiency. However, the metasurfaces based on compact approach are limited in NA preventing them to obtain optimum contribution from larger $k$ vectors.

## Discussion

In conclusion, we have designed and experimentally demonstrated a metasurface spatial filter for all-optical isotropic edge detection in the visible spectral range for real-life scenarios, even with incoherent and unpolarized illumination. Polarization-independent edge detection is demonstrated by simulations at RGB wavelengths and experimentally verified with red and green laser sources. Broadband edge detection along the entire visible range is experimentally verified using an incoherent, unpolarized, white-light source mimicking a realistic scenario. Moreover, simulations indicate that the same metasurface can perform edge detection over a much broader wavelength range extending to near-IR wavelengths. Successful approximation to 2nd

order differentiation transfer function in near-IR range is also verified from measured spatial transmission profile. We evaluate the efficiency of the metasurface using previously defined metrics and introduce an alternative performance metric. We prove that the Fourier filtering approach based on metasurfaces has significant advantages making it suitable for use in commercial applications. We believe our findings are a significant step towards realization of compact, low-power, and ultrafast all-optical data and image processing systems.

## Methods

### Simulations

For the unit cell and device simulations, commercial simulation program Lumerical FDTD Solutions is used. To represent illumination from the substrate side, plane wave sources with the propagation direction of $+z$ are placed inside the $SiO_2$ substrate. The unit structures and device are placed on top the substrate $(+z)$. The transmitted fields are recorded from a field monitor that is placed above the structures. For the unit cell simulations, periodic boundary conditions (BC) is used along $x$ axis and perfectly matched layers (PML) BC is used along $z$ axis. For device simulations, symmetric and antisymmetric BC are used along $x$ and $y$ axes and PML BC is used along $z$ axis. Substrate is embedded into PML boundaries in both cases to model semi-infinite (relative to the structure) substrate thickness.

### Spatial profile measurements

The measurement setup consisted of an inverted microscope coupled to a spectrometer with CCD camera, and a broadband halogen lamp was used as a light source. The transmission measurements were calibrated with respect to free space. For visible wavelengths, Andor Spectrometer with Newton EMCCD is used. For the NIR measurements, SpectraPro 300i spectrometer with NIRvana 640 is used.

### Sample fabrication

The lift-off technique is used for sample fabrication. Fused Silica wafers are used as substrate. The substrate is spin coated with the electron beam resist PMMA 950 A2 with 2000 rpm for 45 s. After baking, they are coated with conducting polymer (DisCharge DI). The device is patterned by electron beam lithography (Ratih EBL 50 kV). After washing out the conductive polymer with deionized water, the patterned surface is developed in IPA/MIBK mix with a ratio of 3 to 1, for 60 s. Afterwards, the developed metasurface is coated with Aluminum using e-beam evaporator (AJA ATC-E) with a deposition rate of 0.5 Å/s. Finally, the patterned resist is lifted-off by acetone inside a sonic bath.

### Reporting summary

Further information on research design is available in the Nature Portfolio Reporting Summary linked to this article.

## Data availability

The data that support the findings of this study are available from the corresponding author upon request.

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

## Acknowledgements

K.A. acknowledges support from the Air Force Office of Scientific Research under award number FA9550-22-1-0300. This work made use of the NUFAB and EPIC facilities of Northwestern University's NUANCE Center, which has received support from the SHyNE Resource (NSF ECCS-2025633), the IIN, and Northwestern's MRSEC program (NSF DMR-1720139). We thank Dr. Serkan Butun for his assistance and invaluable tips on sample fabrication.

## Author contributions

I.T. and K.A. conceived the idea, I.T. designed the devices and conducted the simulations, I.T. and S.A.D. conducted the fabrication, experiments, and measurements. All authors commented on the manuscript and K.A. supervised the project.

## Competing interests

The authors declare no competing interests.
