## [Peer Review File · Nature Communications]

Metasurface Enabled Broadband All Optical Edge Detection in Visible FrequenciesREVIEWER COMMENTS

Reviewer #1 (Remarks to the Author):

This manuscript introduces the conception, simulation, fabrication, and experimental validation of a metasurface functioning alongside a 4f lens system for edge detection. The metasurface can perform edge detection even under conditions of unpolarized, incoherent, and broadband incident light. The data collected from the measurements aligns well with the design expectations.

However, I do not recommend this manuscript for publication. The application of a 4f system for edge detection is a well-established concept in standard textbooks. Moreover, the edge detection effect achieved by this metasurface can be replicated by simply using an annular aperture consisting of a center opaque disk and concentric opaque rings. I think edge detection is still feasible even without the rings as long as the diameter of the center disk is large enough to block the 0th order diffraction at the Fourier plane. If the annular aperture is used in edge detection, it will also possess broadband, incoherent, and polarization-insensitive properties. Therefore, I don't see any novelty in using a nanolithography-fabricated metasurface to achieve what the annular aperture can do. In fact, a closer examination of the amplitude and phase profiles required for the metasurface reveals that the phase profile remains constant, and the amplitude profile changes progressively from 0 to 1 from the center to the edge, which is similar to the optical property of an annular aperture.

Reviewer #2 (Remarks to the Author):

The recent advances on optical metasurfaces lead to development of novel optical device with miniature dimension. Optical imaging is one of the technologies that could take good advantage of the ultrathin optical metasurfaces. Particularly, recent reports show that metasurface could be used for real-time edge detection for advanced image processing. However, there are several practical challenges such as polarization dependence and not operating in broad spectral range.

This manuscript demonstrates, for the first time, the use of optical metasurfaces to enable real time edge detection with polarization independent, broadband and high transmission efficiency. The researchers designed and fabricated the metasurface device based on centric aluminum rings. They confirmed the results for both coherent laser and incoherent light source using 4f imaging and edge-detection setup. They extended the demonstrations for both visible and near-IR range.

The results and experimental techniques reported in the manuscript are novel and clear, and represent a significant advancement in metasurface imaging capability, and its applications for edge detection of meta-imaging devices. Therefore, I would recommend the paper be published in *Nature Communication* with revisions that addressing the following comments/questions.

1. The authors indicate the deviation on the optimal optical response with the experimental result for wavelength of 450 nm. But it is not clear how this deviation affecting the final edge detection capability. Further discussion could be included.
2. The experimental results reported are currently limited from wavelength 450-660 nm and with certain deviation in the short wavelength range. The author should provide discussion on further extending the wavelength range in the visible range. Will dielectric-type metasurface further advances the performance metrics?
3. The authors claim that the performance metrics are generally not well defined (e.g., edge detection efficiency, quality) and there are many papers related to edge-detection based on metasurfaces. More detailed comparison could be provided to enhance the understanding of performance advantages of this work.
4. In Fig. 4c,e,g,f, why the top and bottom parts of the images are cut off?
5. In Fig. 3, the contrast and magnifications of the SEM images are not clear. What are the different between (b) and (c) in term of actual width?
6. How uniform is the 2mm metasurfaces? For examples, how the transmission profile (Fig. 3d) looks like for different azimuthal directions?

Reviewer #3 (Remarks to the Author):

The work proposes a broadband, polarization-independent edge detection device. It is interesting and suitable for publication in this magazine.

The authors chose a structure composed in figure 1a, with parameters given in the caption. How did they recalculate these parameters. And if they can point out a possible optimization procedure of the unit cell.

An approximation is inevitable in the procedure of converting the unit cell from the Cartesian system to the cylindrical one. How this type of approximation was taken into account.

Reviewer #1 (Remarks to the Author):

This manuscript introduces the conception, simulation, fabrication, and experimental validation of a metasurface functioning alongside a 4f lens system for edge detection. The metasurface can perform edge detection even under conditions of unpolarized, incoherent, and broadband incident light. The data collected from the measurements aligns well with the design expectations.

However, I do not recommend this manuscript for publication. The application of a 4f system for edge detection is a well-established concept in standard textbooks. Moreover, the edge detection effect achieved by this metasurface can be replicated by simply using an annular aperture consisting of a center opaque disk and concentric opaque rings. I think edge detection is still feasible even without the rings as long as the diameter of the center disk is large enough to block the 0th order diffraction at the Fourier plane. If the annular aperture is used in edge detection, it will also possess broadband, incoherent, and polarization-insensitive properties. Therefore, I don't see any novelty in using a nanolithography-fabricated metasurface to achieve what the annular aperture can do. In fact, a closer examination of the amplitude and phase profiles required for the metasurface reveals that the phase profile remains constant, and the amplitude profile changes progressively from 0 to 1 from the center to the edge, which is similar to the optical property of an annular aperture.

We respectfully disagree with the reviewer's opinions about the novelty of our work for the following reasons;

- 1) The reviewer states: “The application of a 4f system for edge detection is a well-established concept in standard textbooks.” We agree with that statement. As a matter of fact, we referred to one these standard textbooks (Ref.4) for further explanations of 4f systems in the first paragraph of the introduction at the manuscript;
“... *the Fourier optics principles based on 4f systems, where lenses are used to perform spatial Fourier transformations of the optical fields creating 4f distance between the object and the image [4-6].*”

However, conventional optics are bulky and prone to integration issues. To miniaturize the conventional optical systems and address related issues, metasurfaces emerged as a promising solution. Based on metasurfaces, various optical components including polarizers, holograms, beam deflectors, and lenses have been previously reported. As a part of these miniaturization efforts, metasurface spatial filters have also been demonstrated in recent years (Ref.5-15) as it is discussed in the first paragraph of the introduction of the manuscript as;

“Although conventional bulky optics suffer from large footprint and difficulties in integration, nanophotonic approaches based on metamaterials and metasurfaces have proven to successfully address such problems. In recent years, several approaches based on metamaterials and metasurfaces have been quite successful in performing analog computation, specifically, edge detection tasks. Earlier work put emphasis on the Fourier optics principles based on 4f systems, where lenses are used to perform spatial Fourier

transformations of the optical fields creating $4f$ distance between the object and the image [4-6]. In addition to numerous theoretical proposals [7-12], optical computing based on Pancharatnam-Berry phase [13,14] and tailored resonances [15] have been demonstrated experimentally.”

- 2) The reviewer states that edge detection can be achieved by an annular aperture, which consists of an opaque disk and may or may not include co-centric rings. We partially agree with that statement. Indeed, our structure itself can be defined as “an opaque disk with co-centric rings”, which is quite similar to our description at the end of the page 4 of the manuscript;

“Hence, the resulting device becomes a collection of co-centric rings of Al ridges with varying width, as schematically depicted in Figures 2a,e.”

However, edge detected images by an annular aperture described by the reviewer will be lower quality. To demonstrate the difference, we numerically calculated the output of a $f\bar{f}$ system, where the annular apertures described by the reviewer and a second order differentiator are used as spatial filters. For the calculations, the image size is set to 251x251 pixels. Four different annual apertures are demonstrated as example cases; an opaque disk with 50 pixels diameter, an opaque disk with 200 pixels diameter, an opaque disk with 80 pixels diameter and a 40 pixels wide ring, an opaque disk of 50 pixels diameter inside a 40 pixels wide ring that is inside a 20 pixels wide ring. Here, we assumed the annular aperture feature sizes are much larger than the operation wavelength as the reviewer suggests simple fabrication instead of nanofabrication for such apertures. As a result, the annular aperture is in the ray optics regime. For the annular apertures, the resulting transmission profile becomes a pattern of dark (ring/disk) and bright (everywhere else) regions in spite of our metasurface, which can provide intermediate transmission values due to its sub-wavelength features. Note that we omitted the diffractive annular aperture case since the phase response of such an aperture will not be constant, which contradicts with both the reviewer’s assumption on its properties and our target requirements. The transmission profiles of the 2nd order differentiator and the annual apertures are shown in Figure R1 (a-e). The 2nd order differentiator outperforms the annual apertures as seen in Figure R2 (a-e). The quality difference can also be visualized from the cross sections ($x=0^{\text{th}}$ pixel) provided in Figure R2 (f-h).

We believe that our manuscript is providing solutions to multiple problems associated with the metasurface-based edge detection devices. Reviewer’s suggestions are valid , however as shown with simple numerical calculations, a basic annular aperture will not perform at the same level, and there are significant improvements and advantages in using nanofabricated meta-surface based optical edge detection devices.

Figure R1. The spatial filters and the input field. The transmission profile of the annular apertures; (a) opaque disk with 50 pixel diameter, (b) opaque disk with 200 pixel diameter, (c) opaque disk with 80 pixel diameter and a concentric ring with 40 pixel width, (d) opaque disk with 50 pixel diameter and two concentric rings with 40 pixel width and 20 pixel width, respectively. (e) The transmission profile of a 2nd order differentiator. (f) The input field.

Figure R2. (a-e) The output field of the 4F system corresponding to spatial filter transmission profiles provided in Fig R1 (a-e), respectively. The cross sections along $x=0^{\text{th}}$

pixel line for; (f) 50 pixel diameter opaque disk case, (g) 200 pixel diameter opaque disk case, (h) 2nd order differentiator.

Reviewer #2 (Remarks to the Author):

The recent advances on optical metasurfaces lead to development of novel optical device with miniature dimension. Optical imaging is one of the technologies that could take good advantage of the ultrathin optical metasurfaces. Particularly, recent reports show that metasurface could be used for real-time edge detection for advanced image processing. However, there are several practical challenges such as polarization dependence and not operating in broad spectral range.

This manuscript demonstrates, for the first time, the use of optical metasurfaces to enable real time edge detection with polarization independent, broadband and high transmission efficiency. The researchers designed and fabricated the metasurface device based on centric aluminum rings. They confirmed the results for both coherent laser and incoherent light source using 4f imaging and edge-detection setup. They extended the demonstrations for both visible and near-IR range.

The results and experimental techniques reported in the manuscript are novel and clear, and represent a significant advancement in metasurface imaging capability, and its applications for edge detection of meta-imaging devices. Therefore, I would recommend the paper be published in *Nature Communication* with revisions that addressing the following comments/questions.

We thank very much the Reviewer for the positive and constructive feedback.

1. The authors indicate the deviation on the optimal optical response with the experimental result for wavelength of 450 nm. But it is not clear how this deviation affecting the final edge detection capability. Further discussion could be included.

We thank the reviewer for constructive remarks. As suggested by the reviewer, we added further explanations to the manuscript on the performance of metasurface at 450 nm wavelength. The updated discussions are provided below, where the updates are underlined.

Page 4, last paragraph;

“However, the deviation in transmission becomes substantial at 450 nm (Fig. 1f). Note that the metasurface still maintains a gradually increasing transmission profile from center towards the edge. It suppresses the low-frequency components and operates as a high-pass spatial filter, which is desired for edge detection. As a result, our metasurface is expected to maintain edge detection operation at 450 nm with a relatively lower quality than the design wavelength.”

Page 6, last paragraph;

“Although the response function of the device deviates from the target response at 450 nm (Fig 1f and Fig 2b,f), the metasurface maintains its edge detection ability as it still operates as a high-pass filter blocking low spatial frequencies. However, since its transmission is larger than the target response at low special frequencies (closer to center of device), the background suppression is weaker at 450 nm than 550 and 660 nm wavelengths.”

2. The experimental results reported are currently limited from wavelength 450-660 nm and with certain deviation in the short wavelength range. The author should provide discussion on further extending the wavelength range in the visible range.

To cover the entire visible range, we used the Xe lamp as a white light source to obtain the result presented in the Figure 5 in the manuscript. Xenon lamp's broadband output spectrum is continuous and almost uniform across the visible region, which is explained in the first paragraph of page 9 of manuscript as;

"... we replaced the laser source with a Xenon arc lamp in the 4f system (Figure 4a) since its broadband output spectrum is continuous and almost uniform across the visible region ..."

However, simulated results were limited to 450 nm-660 nm range as pointed out by the reviewer. To further extend this range to cover entire visible spectrum as suggested by the reviewer, we simulated the 3D device at 400 nm and 700 nm as well. The results are added to Supporting Information (new SI-2).

Will dielectric-type metasurface further advances the performance metrics?

Dielectric metasurfaces provides substantial increase in transmission efficiency for applications, such as lensing, beam steering, and polarization conversion, by minimizing the losses. However, we believe achieving even similar performance with dielectric metasurfaces will be much more challenging.

In our case, the target response requires spatially distributed losses in transmission, either by reflection or absorption. Dielectric metasurfaces, in general, lack absorption and rely mainly on reflection as the loss mechanism. As a result, achieving the target behavior will be more challenging with dielectric metasurfaces. For example, the 2nd order differentiator requires zero "0" transmission at its center (See Figure 1(d-f)), which is mainly achieved at strong resonances using dielectric materials. As a result, the operation will be narrowband.

3. The authors claim that the performance metrics are generally not well defined (e.g., edge detection efficiency, quality) and there are many papers related to edge detection based on metasurfaces. More detailed comparison could be provided to enhance the understanding of performance advantages of this work.

We thank the reviewer for constructive suggestions. As stated by the reviewer, there are many papers on metasurface based edge detection. Although, unfortunately, many of these works do not use or report a quantitative efficiency definition and rely on qualitative evaluations for efficiency and performance [Refs 15,16,19,21,24,28], to date, several efficiency and performance metrics are reported in literature [Refs 13,14,22,31]. For example, J. Zhou et al. [Ref. 13] defined the efficiency as $\eta = \frac{P_{LCP} + P_{RCP}}{P_{input}}$, which is inapplicable in our case. In another work, Y. Zhou et al. [Ref. 22] proposed an efficiency definition as the square of the transfer function $|H(k)|^2$ at the maximum spatial frequency that can be fitted to the target mathematical function. However, there is no widely acknowledged performance metric yet, to the best of our knowledge. We pointed out this issue in the last paragraph of page 9 of the manuscript as;

“ To date, various edge-detection efficiency definitions are reported in literature [13,14, 22, 31], and there is not an agreement on the performance metrics...”

In an effort to address this problem, we proposed to use contrast as a performance metric and also used the efficiency values proposed by Cotrufo et al. (Ref. 31) for evaluating performance of our metasurface, which is discussed in page 9-10 of the current manuscript;

“ To date, various edge-detection efficiency definitions are reported in literature [13,14,22,31], and there is not an agreement on the performance metrics. Here, we propose to use contrast as an alternative figure-of-merit, which measures the difference between bright and dark regions in an image...”

To further assess the quality of edge detection, we calculated the efficiency metrics proposed by Cotrufo et. al [31] ... ”

4. In Fig. 4c,e,g,f, why the top and bottom parts of the images are cut off?

In Figure 4, the effect pointed out by the reviewer stems from the experimental problems that we couldn't fully resolve. In these experiments, the input field is not uniform due to beam size and shape. The intensity profile can be approximated as a Gaussian that is highest at the center and decreases towards the edges of the frame. Although the beam can cover the entire amplitude mask, it is attenuated to obtain a clear input field as seen in Figure R3, which demonstrates the bright and attenuated cases for the green laser source and Northwestern University logo (Figure 4e). Although the intensity decrease is not clear at the input images as the camera is already saturated around the center, at the edge detected images, edges further from the center seem darker as seen in Figure R3 c,f. This effect becomes clearer for the Northwestern Wildcats logo as it has a larger open area than the Northwestern University logo (Figure 4).

Figure R3. (a) The bright and (b) attenuated input fields created by a HeNe green laser and (d, e) corresponding edge detected images, respectively. (c, f) Zoom-in to the top of the attenuated input and output images, respectively.

5. In Fig. 3, the contrast and magnifications of the SEM images are not clear. What are the different between (b) and (c) in term of actual width?

SEM images from same area with higher magnification are added to the Supporting Information (SI). The Al ridge widths at Figure 3 (b) and (c) are calculated as ~ 380 nm and ~ 80 nm, respectively based on these SEM images. The updated part of the SI is provided below;

“SI-4 Additional SEM Images

Figure S4 exhibits the higher magnification scanning electron microscope (SEM) images of the areas shown in Figure 3 b,c.

Figure S4. The SEM images of the fabricated metasurface. (a) between the center and the edge of the metasurface (area shown in Figure 3c). (b) Close to the edge of metasurface (area shown in Figure 3b).”

6. How uniform is the 2mm metasurfaces? For examples, how the transmission profile (Fig. 3d) looks like for different azimuthal directions?

Measured transmission profiles for different azimuthal directions are provided in Figure R4 below for the reviewer’s information. Based on these figures and SEM images discussed above (Fig.3 and Fig. S4), we conclude that the 2mm sample is reasonably uniform.

Figure R4. Measured transmission profiles at the design wavelength (660 nm) through various azimuth angles.

Reviewer #3 (Remarks to the Author):

The work proposes a broadband, polarization-independent edge detection device. It is interesting and suitable for publication in this magazine.

We thank very much the Reviewer for the positive remarks.

The authors chose a structure composed in figure 1a, with parameters given in the caption. How did they recalculate these parameters. And if they can point out a possible optimization procedure of the unit cell.

We followed the forward design methodology to determine required unit cell parameters. The unit cell periodicity intuitively chosen as 500 nm since large values may result in additional diffraction orders at short end of the visible range and small values limits would result in an inadequate range of Al ridge width to effectively approximate the target response. The periodicity would be subjected to iterations if 500 nm periodicity didn't work. The height is chosen as 70 nm after a few iterations of continuous Al film with varying thickness, where we aimed to find the thickness that do not transmit light. After fixing the periodicity and thickness, we created a library of unit cells by sweeping the width of the Al ridges as shortly discussed in the beginning of the page 4 of the manuscript;

“To realize the desired optical response, we created a library of 2D unit cells with a fixed periodicity of 500nm. The unit cells consist of Aluminum wires with a constant height and varying widths ranging from 0 to 500 nm on top of a SiO₂ substrate (Figure 1a). A width of 0 nm corresponds to a bare substrate, while 500 nm represents a continuous film. The height of the metal stripe is set to 70 nm to ensure no transmission in the 500 nm width (continuous film case). The minimum stripe width (excluding the no stripe case) was set to 50 nm due to fabrication constraints.

”

An approximation is inevitable in the procedure of converting the unit cell from the Cartesian system to the cylindrical one. How this type of approximation was taken into account.

The 2D device was on the xz plane with the implying that there is no variation along y axis i.e., $y \rightarrow \infty$, and x polarized plane wave illumination was considered during the simulations. In this case, the y component is the in-plane orthogonal (to the polarization of incident wave) component of the structure. Thus, we can project it to azimuth “ Φ ” in cylindrical coordinates, which will be in plane orthogonal dimension under any arbitrary polarization of incidence. As a result, $y \rightarrow \infty$ assumption maps to $\Phi \rightarrow 2\pi$. Similarly, the in plane parallel component “ x ” is projected to radial distance “ r ”. As the direction light propagation is the same, the z axis was not changed. The conversion and approximations are briefly explained in the last paragraph of page 4 in the manuscript;

“ We map the 2D hypothetical device to 3D by converting position to radial distance ($x \rightarrow r$; $r = \sqrt{x^2 + y^2}$). This is basically a coordinate system transformation from cartesian to cylindrical coordinates. In this case, the underlying infinity assumption for the in-plane orthogonal dimension of the unit structure, i.e., stripe length in $y \rightarrow \infty$, in 2D simulations maps to the Φ (azimuthal) component for the 3D device. Since the azimuthal axis has a range from 0 to 2π , the stripe length becomes 2π in Φ corresponding to a full circle. Hence, the resulting device becomes a collection of co-centric rings of Al ridges with varying width...”

REVIEWER COMMENTS

Reviewer #1 (Remarks to the Author):

Upon reviewing the authors' first response to my comments, they noted that, "However, conventional optics are bulky and prone to integration issues. To miniaturize the conventional optical systems and address related issues, metasurfaces have emerged as a promising solution...". I would like to reiterate, however, that the authors' work does not, in fact, result in any miniaturization. The presented setup is still dependent on a 4f system (which enables the broadband operation) that necessitates the inclusion of two conventional refractive lenses, which results in most of the volume (due to focal length) and weight.

In response to my original comment concerning the use of an annular aperture for edge detection, I am grateful for the added simulations. However, how were the numbers, positions and sizes of rings determined? It seems the authors only randomly picked a few different cases without any rigorous consideration. It is very likely a more thorough optimization process could significantly enhance the performance of the annular aperture, possibly even equating it to the resolution presented by the quadratic-amplitude AI metasurface.

Equally perplexing is the simulation data which appears to contain several anomalies, as indicated in the figure below, adopted from the authors' Fig. R2b. The authors have not provided any explanation on how this simulation was conducted. Consequently, I am left to presume that it was carried out based on Fourier transform considering an ideal (aberration-free) 4f system. Given this assumption, the regions corresponding to the same radius (highlighted by the yellow dashed circle) should present the same resolution. However, one can see the regions, for instance, depicted by the blue and green arrows possess different blurriness. Furthermore, it's quite perplexing to observe that the blurred regions, as indicated by the red arrows, align strictly along either vertical or horizontal trajectories. This outcome should not happen in a lens system demonstrating rotational symmetry.

Taking into account the above-mentioned points, my initial recommendation to reject the manuscript remains unchanged.

Reviewer #2 (Remarks to the Author):

The authors addressed most of my questions and concerns. I recommend it to be published in Nature Communication.

Reviewer #3 (Remarks to the Author):

The authors have responded to my questions. Therefore, the paper is suitable for publication.

Reviewer #1 (Remarks to the Author):

Upon reviewing the authors' first response to my comments, they noted that, "However, conventional optics are bulky and prone to integration issues. To miniaturize the conventional optical systems and address related issues, metasurfaces have emerged as a promising solution...". I would like to reiterate, however, that the authors' work does not, in fact, result in any miniaturization. The presented setup is still dependent on a 4f system (which enables the broadband operation) that necessitates the inclusion of two conventional refractive lenses, which results in most of the volume (due to focal length) and weight.

We did not claim complete miniaturization in our manuscript and agree with reviewer's comment that a 4f system requires refractive lenses. However, we would like to draw reviewer's attention to recent miniaturization efforts in the literature cited in our manuscript [Refs 7-15 of manuscript] using metasurface based spatial filters to reduce the footprint of one critical optical element therefore contributing to overall reduced size of the entire system .

We would like to highlight the fact that our metasurface is compatible for integration with metalenses to create a complete miniature system that cannot be achieved with conventional bulky apertures. Even metalenses were integrated with a commercial aperture to create a 4F system, the system will remain bulky due to the aperture size. Apertures can be further minimized using similar lithography and deposition techniques for our metasurface approach, which were criticized by the reviewer themselves in the first round of revision.

It is important to stress a point that might be overlooked in reviewer's comments, which is the miniaturization is not just achieved with reduced thickness but also overall reduction in the diameter of the components. In this work, we demonstrated by simulation that our metasurface can be designed in diameters in μm range. In that size range, the conventional apertures suggested by the reviewer will suffer from diffraction effects [1]. As a result, such an aperture cannot provide the required spatial transfer function. To prove our hypothesis, we simulated the far-field response of an aperture with $30\ \mu\text{m}$ diameter and an area of interest (or system size) with $50\ \mu\text{m}$ edge length. As seen from figure R1, in this size range, far-field response of the aperture is different from geometric optics expectations and does not provide the required spatial transfer function. In conclusion, a simple aperture cannot be used to achieve edge detection in this size range. To

achieve such minimization, one need to carefully design a structure depending on either diffractive optics or metasurface principals.

Figure R1. The electric field magnitude of the 30 μm aperture at far-fields ($z= 100 \mu\text{m}$).

Simulation details: To model the aperture, we used a perfect electrical conductor disk with 70 nm thickness (same thickness of our unit structure) on top of a low index substrate. A plane wave source is placed inside the substrate and near-fields are collected by a field monitor above the disk and far-field response is calculated by the built-in function “farfieldexact”. The simulations were performed for 660 nm wavelength. The simulation is performed using the commercial simulation tool, Ansys Lumerical FDTD.

[1] Wolfe, P. Diffraction of a Plane Wave by a Circular Disk. *J. Math. Anal. Appl.* **1979**, 67 (1), 35–57. [https://doi.org/https://doi.org/10.1016/0022-247X\(79\)90005-2](https://doi.org/https://doi.org/10.1016/0022-247X(79)90005-2).

In response to my original comment concerning the use of an annular aperture for edge detection, I am grateful for the added simulations. However, how were the numbers, positions and sizes of rings determined? It seems the authors only randomly picked a few different cases without any rigorous consideration. It is very likely a more thorough optimization process could significantly enhance the performance of the annular aperture, possibly even equating it to the resolution presented by the quadratic-amplitude AI metasurface.

It seems like the reviewer might have forgotten their initial comment in the first round of revision suggesting that a conventional aperture can achieve edge detection with similar performance. In order to address their concern, we followed reviewer’s suggestion: *“Moreover, the edge detection effect achieved by this metasurface can be replicated by simply using an annular aperture consisting of a center opaque disk and concentric opaque rings. I think edge detection is still feasible even without the rings as long as the diameter of the center disk is large enough to block the 0th order diffraction at the Fourier plane.”*

We showed that the output quality will be lower as a result of the difference of the transfer functions.. We believe that we addressed reviewer's initial comments appropriately. At this point of revision, further optimization about reviewer's initial idea is beyond the scope of this work.

Equally perplexing is the simulation data which appears to contain several anomalies, as indicated in the figure below, adopted from the authors' Fig. R2b. The authors have not provided any explanation on how this simulation was conducted. Consequently, I am left to presume that it was carried out based on Fourier transform considering an ideal (aberration-free) 4f system. Given this assumption, the regions corresponding to the same radius (highlighted by the yellow dashed circle) should present the same resolution. However, one can see the regions, for instance, depicted by the blue and green arrows possess different blurriness. Furthermore, it's quite perplexing to observe that the blurred regions, as indicated by the red arrows, align strictly along either vertical or horizontal trajectories. This outcome should not happen in a lens system demonstrating rotational symmetry.

We thank the reviewer for carefully inspecting our simulation results that are reported in our response to their initial comments. The data is created to indicate the differences of the transfer functions (differentiator and aperture) apart from the other limitations. Thus, to avoid other sources of errors, we calculated system output assuming ideal fourier transformation as correctly noted by the reviewer. The remaining error sources are due to digitization and numeric approximation errors that are negligible. We provide our numerical codes with this revision so the reviewer can check and ensure that there is no data manipulation.

The difference in blurriness results from the interference effect due to the non-ideal transfer function of the aperture. Since the input field is not symmetric, the blurriness is not symmetric as well.

Taking into account the above-mentioned points, my initial recommendation to reject the manuscript remains unchanged.

We believe that we addressed reviewer's initial concerns and remaining concerns are also addressed with this response. Given positive reviews from remaining two reviewers, we believe that our manuscript should be considered for publication.

Reviewer #2 (Remarks to the Author):

The authors addressed most of my questions and concerns. I recommend it to be published in Nature Communication.

We thank very much the Reviewer for their time and constructive feedback throughout the revision process.

Reviewer #3 (Remarks to the Author):

The authors have responded to my questions. Therefore, the paper is suitable for publication.

We thank very much the Reviewer for their time and constructive feedback throughout the revision process.

REVIEWERS' COMMENTS

Reviewer #1 (Remarks to the Author):

I'd like to emphasize upfront that this article doesn't meet the novelty criteria required for publication in Nature Communications. Recent works on the use of metasurfaces for edge detection, which are notable because of their angle-sensitive transmission and non-local properties. As such, these metasurfaces can achieve edge detection without the conventional 4f-lens system, leading to reductions in system weight, volume, and potentially cost.

The bulk of this manuscript revisits well-established knowledge. The need for the transmission as a quadratic function of spatial coordinates is well-documented in the cited references and even in Goodman's textbook of Fourier Optics. Additionally, the methodology employed by the authors, involving the 4f-lens system, is a textbook example of performing edge detection in Fourier Optics.

The primary contribution of this manuscript appears to be the design and realization of the AI metasurface. While this may be worth publication in a technically-oriented journal, it doesn't align with the standards of Nature Communications.

One point of interest the authors have raised during the peer-review duration is the claim that the AI metasurface's edge detection capability surpasses that of the traditional 4f-system. Should the authors offer robust evidence supporting this claim, I'd be inclined to recommend the article for publication. However, it seems the revisions have not adequately addressed this, perhaps because I am the sole reviewer who is against the work's publication.

In conclusion, I cannot recommend the publication of this metasurface work for edge detection that relies on the 4f-system; the technique isn't novel. If the authors could deliver a compelling and fair comparison demonstrating superior performance compared to traditional 4f-based edge detection in the main text, I'd reconsider my recommendation.

Reviewer #1 (Remarks to the Author):

I'd like to emphasize upfront that this article doesn't meet the novelty criteria required for publication in Nature Communications. Recent works on the use of metasurfaces for edge detection, which are notable because of their angle-sensitive transmission and non-local properties. As such, these metasurfaces can achieve edge detection without the conventional 4f-lens system, leading to reductions in system weight, volume, and potentially cost.

The bulk of this manuscript revisits well-established knowledge. The need for the transmission as a quadratic function of spatial coordinates to a is well-documented in the cited references and even in Goodman's textbook of Fourier Optics. Additionally, the methodology employed by the authors, involving the 4f-lens system, is a textbook example of performing edge detection in Fourier Optics.

The primary contribution of this manuscript appears to be the design and realization of the AI metasurface. While this may be worth publication in a technically-oriented journal, it doesn't align with the standards of Nature Communications.

One point of interest the authors have raised during the peer-review duration is the claim that the AI metasurface's edge detection capability surpasses that of the traditional 4f-system. Should the authors offer robust evidence supporting this claim, I'd be inclined to recommend the article for publication. However, it seems the revisions have not adequately addressed this, perhaps because I am the sole reviewer who is against the work's publication.

In conclusion, I cannot recommend the publication of this metasurface work for edge detection that relies on the 4f-system; the technique isn't novel. If the authors could deliver a compelling and fair comparison demonstrating superior performance compared to traditional 4f-based edge detection in the main text, I'd reconsider my recommendation.

As requested by the reviewer, we included the comparison of traditional high-pass filter (HPF) approach and 2nd order differentiation. In addition to the results that we shared with the reviewer previously, we also calculated the signal-to-noise ratio (SNR) of the output images to provide a quantitative comparison. The SNR values are calculated as 12.8 dB, 14.5 dB, and 26.3 dB, respectively for 50-pixel HPF, 200-pixel HPF, and 2nd order differentiator. The comparison indicates that our approach surpasses the traditional HPF filtering with opaque disks.

The corresponding discussion is added to supporting document and provided here for the reviewer's reference;

“

SI-1 Comparison of 2nd Order Differentiation and High-pass Filtering

We numerically calculated the output of a 4f system with conventional sharp high-pass filters (HPFs) and a second order differentiator as spatial filters. To indicate the differences of the transfer functions apart from the other limitations and to avoid other sources of errors, we calculated system output using fast-fourier transformation. For the calculations, the image size is set to 251x251 pixels.

Two different HPFs are demonstrated as example cases; an opaque disk with 50 pixels diameter and an opaque disk with 200 pixels diameter as seen in Figure S1. Here, we assumed the spatial feature sizes are much larger than the operation wavelength as the required transfer function can be easily obtained using opaque disks in ray optics regime, yet they fail when the diffraction effects take over in smaller dimensions [1]. For the HPFs, the resulting transmission profile becomes a pattern of dark (disk) and bright (everywhere else) regions in spite of our metasurface, which can provide intermediate transmission values due to its sub-wavelength features.

To quantitatively compare the edge detection quality, we calculated the signal-to-noise ratios (SNR) of the output images shown in Figure S2. The SNR is calculated in dBs as,

$$SNR = 10\log_{10} \left(\frac{\sum \sum \frac{r(x,y)^2}{[r(x,y)-s(x,y)]^2} \right) \quad (1)$$

where $s(x,y)$ is the output image and $r(x,y)$ is only the edges at the output image. The resulted values are 12.8 dB, 14.5 dB, and 26.3 dB, respectively for 50 pixel HPF, 200 pixel HPF, and 2nd order differentiator. The quality difference can also be visually inspected and qualitatively observed from the images and their cross sections ($x=0^{\text{th}}$ pixel) provided in Figure S2.

Figure S1. The spatial filters and the input field. The transmission profile of the annular apertures; (a) opaque disk with 50 pixel diameter, (b) opaque disk with 200 pixel diameter, (c) 2nd order differentiator. (d) The input field.

Figure S2. (a-c) The output field of the 4F system corresponding to spatial filter transmission profiles provided in Fig S1 (a-c), respectively. The cross sections along $x=0^{\text{th}}$ pixel line for; (d) 50 pixel diameter opaque disk case, (e) 200 pixel diameter opaque disk case, (f) 2nd order differentiator.

...

References

- (1) Wolfe, P. Diffraction of a Plane Wave by a Circular Disk. *J. Math. Anal. Appl.* **1979**, 67 (1), 35–57.

..